# Active Learning for Molecular Conformation Optimization with a Domain-Agnostic Neural Surrogate Oracle

## Abstract

Molecular conformation optimization is crucial to computer-aided drug discovery and materials design, yet conventional force-based minimization with physics oracles (e.g., DFT) is prohibitively expensive. Neural network potentials (NNPs) are capable of accelerating this process but typically require large quantum chemical datasets for training. To reduce data requirements, active learning (AL) approaches have been designed for this task. The state-of-the-art approach, GOLF, relies on the surrogate oracle to sample new data. However, the surrogate oracle utilizes empirical molecular force fields, which may be absent for a current domain, and thus necessitates careful tuning. We introduce a new AL method for efficient conformation optimization that removes the dependency on empirical force fields. Our approach maintains two NNPs: an online NNP that performs conformation optimization and a target NNP that serves as a trainable surrogate oracle. The target network is an exponential moving average of the online network. During active sampling, the target NNP supplies potential energy estimates that guide data acquisition, while periodic queries to the physics oracle provide ground-truth corrections. Unlike other AL approaches, our method does not require architectural changes to NNP and adds minimal computational overhead compared to the single-model AL pipelines. Across two challenging conformation-optimization benchmarks (based on SPICE2.0 and $\nabla^2 DFT$ datasets) spanning different DFT levels, our method consistently outperforms a baseline NNP trained without AL, achieving substantial improvements with only 1,000 additional conformations.

## 1 Introduction

Molecular conformational optimization is a fundamental task in drug discovery and materials design that is used as a preprocessing step in various quantum chemistry (QC) pipelines (Pracht et al., 2020; Bursch et al., 2022). Conformational optimization is traditionally performed by iteratively minimizing the potential energy $E$ using the interatomic forces $\boldsymbol{F}$ as anti-gradients until the local minima of the potential energy surface (PES) is reached. This process involves multiple expensive QC calculations with a physical oracle (in this work, we also call it a genuine oracle or $\mathcal{O}_G$) for energy and interatomic forces evaluation. A popular choice is to use the density functional theory-based genuine oracle. Notably, the computational cost of density functional theory (DFT) calculations scales at least cubically w.r.t. the number of electrons in the system, which limits its applicability to at best systems with several thousand electrons. Recently, deep neural network potentials (NNPs) (Khrabrov et al., 2024; Batatia et al., 2022; Gasteiger et al., 2021) have emerged as a promising alternative that can accurately approximate DFT-level energies and forces orders of magnitude faster.

Despite being trained on large quantities of QC data, NNPs are still prone to optimization problems such as unstable optimization or convergence to poor local minima: in practice, models trained on popular datasets can encounter a distribution shift during iterative structure relaxation (Tsypin et al., 2024; Khrabrov et al., 2024). The straightforward way to deal with this issue is to enrich the training dataset with optimization trajectories obtained with costly physical oracle calculations (Tsypin et al., 2024; Khrabrov et al., 2024; Fu et al., 2025). Although effective, this approach requires a lot of additional computations.

To overcome this issue, active learning methods are applied to efficiently collect additional data. The core idea behind this family of approaches is to identify a small subset of conformations where the model is most uncertain or likely to be wrong and add them to the training set of the NNP after evaluation with a physical oracle. Tsypin et al. (2024) demonstrated that the NNP can match the quality of the DFT-based oracle in the optimization task, given that the training set was enriched with a sufficient number of conformations from the optimization trajectories. They then proposed an active learning approach for efficient training of NNPs called GOLF, whose key idea is to use a cheap but inaccurate surrogate oracle $\mathcal{O}_S$, that is correlated with the genuine oracle $\mathcal{O}_G$, to decide which conformations are evaluated with $\mathcal{O}_G$ and added to the training dataset. Tsypin et al. (2024) considered molecules in vacuum and showed that utilizing the empirical MMFF94 (Halgren, 1996) force field as a surrogate oracle allows for successfully training an NNP that matches the quality of the DFT-based oracle while cutting down the required additional data by 50 times.

Although the active learning learning scheme proposed in GOLF proved to be efficient for the considered domain, a proper surrogate oracle must be carefully selected for each new domain. Moreover, for some complex domains, such as protein-ligand pairs (Faver et al., 2011; Yilmazer & Korth, 2013), organic liquids (Kovács et al., 2025; Kaminski & Jorgensen, 1996), or crystals (Kriz et al., 2023; Al Mamun et al., 2023; Zuo et al., 2020), it is challenging to find an empirical force field that correlates well with DFT-based methods or experimental results. This motivated us to explore active learning approaches that do not rely on empirical force fields. One common approach is to train an ensemble of NNP models and use their prediction variance as an uncertainty measure (Smith et al., 2018; Kulichenko et al., 2023; Carrete et al., 2023; Tan et al., 2023). While effective, ensemble methods significantly increase computational cost since multiple networks must be trained and evaluated in parallel. Alternatively, a single model can be equipped to predict its own uncertainty by modeling a distribution (mean and variance) over the target value instead of a point estimate (Tan et al., 2023; Xu et al., 2024). Such mean-variance estimation (MVE) networks provide uncertainty without multiple models, but they require architectural modifications that necessitate retraining the model to incorporate uncertainty estimation.

In this work, we propose a new approach to address the challenges with previous active learning approaches by introducing Neural Oracle. The core idea is to replace GOLF's fixed surrogate oracle with a learned surrogate oracle that is updated in tandem with the NNP. Our method draws inspiration from the self-supervised learning BYOL framework (Grill et al., 2020). Similar to BYOL's use of an online and target network, we maintain a secondary neural network (Neural Oracle) that is updated via a slow-moving average of the main model's parameters. The Neural Oracle predicts energies and forces in a way that gradually aligns with the online NNP, without requiring direct supervision. This design provides a stable target for the NNP to compare against, effectively quantifying the model's uncertainty. Unlike GOLF, which relies on an empirical surrogate, our Neural Oracle is adaptable to any chemical domain that can be processed by the NNP. What is more, our approach does not bring additional computational burden caused by training an ensemble of models and does not require any architectural changes or retraining the baseline NNP model.

We demonstrate the efficiency of the proposed GOLF Neural Oracle on a challenging $\nabla^2$DFT conformational optimization benchmark (Khrabrov et al., 2024), where it surpasses all baselines, including the state-of-the-art GOLF active learning framework. Notably, this superior performance is achieved in a highly data-efficient regime, utilizing only 1000 additional conformations. We then apply the best model to a significantly more challenging SPICE2.0 (Eastman et al., 2024) dataset We then apply the best model to a significantly more challenging SPICE2.0 (Eastman et al., 2024) dataset. The SPICE2.0 not only contains data from various chemical domains, including dipeptides, solvated molecules, amino-acid ligand pairs, and water clusters, but also features a more advanced level of DFT theory: $\omega$B97M-D3(BJ)/def2-TZVPPD. The triple basis makes the $\mathcal{O}_G$ calculations very costly, so we test our proposed approach in a data-efficient regime. We show that only 1000 additional conformations are enough to improve upon an already strong non-active learning baseline.

## 2 RELATED WORK

Molecular conformation optimization has been approached using machine learning methods, broadly categorized into models solving position regression tasks, generative methods, and direct optimization strategies.

**Conformation generation** Generative and position regression models aim to directly produce low-energy 3D conformers. Pioneering generative models often operated on internal coordinates, with early works utilizing normalizing flows (Xu et al., 2021) and torsional diffusion (Jing et al., 2022). Since diffusion models have become the dominant generative paradigm, now often operating directly in Cartesian coordinate space (Lee et al., 2024; Nikitin et al., 2025; Liu et al., 2025). Other frameworks like variational approximations and GFlowNets have also been actively explored (Volokhova et al., 2024) Another class of models frames the problem as a direct, one or few-shot regression task. These methods learn an end-to-end mapping from a given molecular representation to its ground-state 3D coordinates from a 2D molecular graph (Xu et al., 2023; Kim et al., 2025), or by refining a low-quality 3D structure (Lu et al., 2024; Wang et al., 2025). However, a common limitation of these generative and direct prediction models is that their outputs are not guaranteed to reside at a local minimum on the potential energy surface. Consequently, a subsequent relaxation using a physically-informed optimizer is typically necessary, motivating the development of iterative optimization strategies.

**Optimization with NNPs** NNPs Khrabrov et al. (2024); Batatia et al. (2022); Gasteiger et al. (2021) offer a computationally efficient alternative to traditional quantum mechanical methods for predicting molecular energies and forces. These potentials can then be used in conjunction with standard optimization algorithms, like BFGS, to perform geometry optimizations at a fraction of the computational cost (Liu et al., 2022; Tayfuroglu et al., 2025; Hao et al., 2022). The quality of NNP-based optimizations can be further improved by incorporating additional conformations into the training data, which helps to alleviate issues arising from distribution shifts (Tsypin et al., 2024; Khrabrov et al., 2024; Fu et al., 2025). Reinforcement learning has been employed to accelerate the optimization algorithm itself (Ahuja et al., 2021; Zamaraeva et al., 2025). In this work, we focus on improving NNPs optimization accuracy without extensive data generation, a problem that active learning is specifically designed to address.

**Active learning for NNPs** Active learning strategies, where the model requests additional calculations for uncertain regions of the conformational space, have also been employed to reduce the amount of data required to train accurate NNPs (Zhang et al., 2019; Kahle & Zipoli, 2022; Bilbrey et al., 2025; Mazitov et al., 2025). A "go-to" approach to uncertainty quantification (UQ) is to use the standard deviation of predictions from an ensemble of models. While generally robust, training and running ensembles incur a significant computational overhead (Smith et al., 2018; Kulichenko et al., 2023; Carrete et al., 2023; Bilbrey et al., 2025; Zhang et al., 2020; Schran et al., 2020). Consequently, substantial research has focused on developing cheaper, single-model UQ methods, such as mean-variance estimation (MVE) (Carrete et al., 2023; Xu et al., 2024), evidential regression (Amini et al., 2020), and Gaussian Mixture Models (GMMs) (Zhu et al., 2023). However, comprehensive benchmarks have shown that despite being faster, single-model methods do not consistently outperform the robustness of ensembles (Tan et al., 2023). Other uncertainty metrics, such as latent space distances, have also been explored (Musielewicz et al., 2024).

Furthermore, several works have specifically tailored active learning approaches for explicit search of transition states in reactive systems (Yang et al., 2021; Price et al., 2025) and the direct conformational optimization of both molecules and materials (Hessmann et al., 2025; Tsypin et al., 2024; Singh et al., 2024; Wang et al., 2024; Shuaibi et al., 2020). Compared to previous works, our research develops an active learning framework that is domain-agnostic, can be applied without architectural modification to the underlying NNP, and crucially minimizes the computational overhead of both model retraining and new training data generation.

## 3 Notation and Preliminaries

We define the conformation $s = \{z, X\}$ of the molecule as a pair of atomic numbers $z = \{z_1, \ldots, z_n\}, z_i \in \mathbb{N}$ and atomic coordinates $X = \{x_1, \ldots, x_n\}, x_i \in \mathbb{R}^3$, where $n$ is the number of atoms in the molecule. We define the oracle $\mathcal{O}$ as a function that takes conformation $s$ as an input and outputs its potential energy $E_s^{\text{oracle}} \in \mathbb{R}$ and interatomic forces $F_s^{\text{oracle}} \in \mathbb{R}^{n \times 3} : E_s^{\text{oracle}}, F_s^{\text{oracle}} = \mathcal{O}(s)$. To denote the ground truth interatomic force acting on the $i$-th atom, we use $F_{s,i}^{\text{oracle}}$. For example, we denote the Psi4-calculated energy as $E_s^{\text{DFT}}$.

## 3.1 NEURAL NETWORK POTENTIALS

In this work, we use equivariant DimeNet++ (Gasteiger et al., 2020) and GemNet-OC (Gasteiger et al., 2022) due to their strong performance in the $\nabla^2$DFT optimization benchmark (Khrabrov et al., 2024). We denote the NNP parametrized by weights $\boldsymbol{\theta}$ that predicts potential energy as $f_{\boldsymbol{\theta}}(s)$ : $\{\boldsymbol{z}, \boldsymbol{X}\} \to \mathbb{R}$. The forces are derived from the predicted energies by taking a partial derivative:

$$\boldsymbol{F}_{\boldsymbol{\theta}}^i(s) = -\frac{\partial f_{\boldsymbol{\theta}}(s)}{\partial \boldsymbol{x}_i}, \tag{1}$$

where $\boldsymbol{F}_{\boldsymbol{\theta}}^i(s) \in \mathbb{R}^3$ is the force acting on the $i$-th atom as predicted by the NNP. We follow (Khrabrov et al., 2024) and use slightly different loss functions for DimeNet++ and GemNet-OC:

$$\mathcal{L}^{\text{DimeNet++}}(s, \mathcal{B}; \boldsymbol{\theta}) = \frac{\rho_1}{|\mathcal{B}|} \sum_{s \in \mathcal{B}} \|E_s - f_{\boldsymbol{\theta}}(s)\|^2 + \frac{\rho_2}{|\mathcal{B}|} \sum_{s \in \mathcal{B}} \frac{1}{n_s} \sum_{i=1}^{n_s} \left\| \boldsymbol{F}_s^i - \boldsymbol{F}_{\boldsymbol{\theta}}^i(s) \right\|^2; \tag{2}$$

$$\mathcal{L}^{\text{GemNet-OC}}(s, \mathcal{B}; \boldsymbol{\theta}) = \frac{\rho_1}{|\mathcal{B}|} \sum_{s \in \mathcal{B}} |E_s - f_{\boldsymbol{\theta}}(s)| + \frac{\rho_2}{\sum_{s \in \mathcal{B}} n_s} \sum_{s \in \mathcal{B}} \sum_{i=1}^{n_s} \left\| \boldsymbol{F}_s^i - \boldsymbol{F}_{\boldsymbol{\theta}}^i(s) \right\|^2, \tag{3}$$

where $\mathcal{B}$ is a batch of conformations, $E_s$ and $\boldsymbol{F}_s$ are the ground truth energies and forces, $\rho_1$ and $\rho_2$ are energy and forces scale coefficients respectively, and $n_s$ is the number of atoms in $s$.

To perform geometry optimization with an NNP, the optimizer **Opt** utilizes the forces $\boldsymbol{F}_{\boldsymbol{\theta}}(s) \in \mathbb{R}^{n \times 3}$:

$$s_{\text{cur}} = s_{\text{prev}} + \textbf{Opt}(\boldsymbol{F}_{\boldsymbol{\theta}}(s_{\text{prev}})). \tag{4}$$

## 3.2 EVALUATION METRICS

To evaluate the quality of optimization with NNPs, we optimize conformations from the test dataset $\mathcal{D}_{test}$ until convergence or the step limit is reached. Each conformation $s$ in the test dataset comes with a ground truth optimal conformation $s_{\textbf{opt}}$ and its energy $E_{s_{\textbf{opt}}}^{\text{DFT}}$, calculated by performing relaxation with $\mathcal{O}_G$. Following (Tsypin et al., 2024), the quality of the NNP-optimization is evaluated with the averaged percentage of minimized energy:

$$\overline{\text{pct}}_{\text{final}} = 100\% * \frac{1}{|\mathcal{D}_{\text{test}}|} \sum_{s \in \mathcal{D}_{\text{test}}} \frac{E_s^{\text{DFT}} - E_{s_{\text{final}}}^{\text{DFT}}}{E_s^{\text{DFT}} - E_{s_{\textbf{opt}}}^{\text{DFT}}} = 100\% * \frac{1}{|\mathcal{D}_{\text{test}}|} \sum_{s \in \mathcal{D}_{\text{test}}} \text{pct}(s_t), \tag{5}$$

where $s_{\text{final}}$ is the final state of the NNP optimization trajectory. Another metric is the residual energy in state $s_{\text{final}}$: $E^{\text{res}}(s_{\text{final}})$. It is calculated as the delta between $E_{s_{\text{final}}}^{\text{DFT}}$ and the optimal energy:

$$E^{\text{res}}(s_t) = E_{s_t}^{\text{DFT}} - E_{s_{\textbf{opt}}}^{\text{DFT}}; \tag{6}$$

Similar to $\overline{\text{pct}}_{\text{final}}$, this metric can also be aggregated over the evaluation dataset:

$$\overline{E^{\text{res}}}_{\text{final}} = \frac{1}{|\mathcal{D}_{\text{test}}|} \sum_{s \in \mathcal{D}_{\text{test}}} E^{\text{res}}(s_{\text{final}}). \tag{7}$$

Generally accepted chemical precision is 1 kcal/mol (Helgaker et al., 2004). Thus, another important metric is the percentage of conformations for which the residual energy is less than chemical precision. We consider optimizations with such residual energies successful:

$$\text{pct}_{\text{success}} = \frac{1}{|\mathcal{D}_{\text{test}}|} \sum_{s \in \mathcal{D}_{\text{test}}} I\left[E^{\text{res}}(s_{\text{final}}) < 1\right]. \tag{8}$$

Lastly, we track the percentage of "diverged" optimizations $\text{pct}_{\text{div}}$. The optimization is considered diverged if the $\mathcal{O}_G$ was not able to calculate the energy of $s_{\textbf{opt}}$ or when the resulting energy $E_{s_{\text{final}}}^{\text{DFT}}$ is larger than the initial energy $E_s^{\text{DFT}}$. We treat the diverged optimization as unsuccessful when calculating the $\text{pct}_{\text{success}}$.

### 3.3 ACTIVE LEARNING

The core idea of active learning is the iterative refinement of the training set by incorporating new data from regions of configuration space where the model is most uncertain, followed by retraining the model on the augmented dataset. The search for such conformations can be performed using molecular dynamics (Kulichenko et al., 2023)) or biased molecular dynamics (Laio & Gervasio, 2008; Sutto et al., 2012). Two popular approaches for UQ are ensemble- and single model-based (Seung et al., 1992; Nix & Weigend, 1994; Schran et al., 2020; Lakshminarayanan et al., 2017).

An ensemble of models is composed of several NNPs that differ slightly (see Section 5 for details). When the predictions of all members are in close agreement, it indicates that the conformation lies within a region of configuration space that was well represented in the training set. In contrast, large discrepancies among the ensemble predictions signal an out-of-distribution conformation. The uncertainty is calculated as (Tan et al., 2023):

$$\rho_{\text{ensemble}}(s) = \frac{\sigma_E(s)}{\sqrt{n_s}}, \qquad \sigma_E^2(s) = \frac{1}{\mathcal{M}-1} \sum_{i=1}^{\mathcal{M}} \big(\tilde{E}_i(s) - \bar{E}(s)\big). \tag{9}$$

Here, $\tilde{E}_i = f_{\boldsymbol{\theta}_i}(s)$ is the energy predicted by the $i$-th ensemble member, and $\bar{E}(s) = \sum_{i=1}^{\mathcal{M}} \tilde{E}_i(s)$ is the ensemble-averaged energy. The variance $\sigma_E(s)$ is normalized with the square root of the number of atoms $n_s$

One widely adopted single-model approach is the mean variance estimation technique (MVE), where training data is treated as Gaussian random variables, and the NNP is trained to predict mean and variance $\mathcal{N}(\mu, \sigma)$. Training is performed via maximum likelihood estimation (Tan et al., 2023):

$$\mathcal{L}^{\text{MVE}}(s, \mathcal{B}; \boldsymbol{\theta}) = \frac{\rho_1}{|\mathcal{B}|} \sum_{s \in \mathcal{B}} \|E_s - f_{\boldsymbol{\theta}}(s)\|^2 + \frac{\rho_2}{|\mathcal{B}|} \sum_{s \in \mathcal{B}} \frac{1}{2n_s} \sum_{i=1}^{n_s} \log\big(2\pi\sigma_{F_i}^2\big) + \frac{\left\|\boldsymbol{F}_s^i - \boldsymbol{F}_{\boldsymbol{\theta}}^i(s)\right\|^2}{\sigma_{F_i}^2}, \tag{10}$$

where $\sigma_{F_i}^2$ denotes the predicted per-atom force variance. For MVE-based models, the uncertainty estimate is computed as:

$$\rho_{\text{MVE}} = \max_{i \in s} \sigma_{F_i}^2. \tag{11}$$

### 3.4 GOLF

The GOLF framework (Tsypin et al., 2024) features a unique data sampling scheme, where the conformation selection is not based on the model's uncertainty. Instead, a conformation is added to the training dataset if the energy (according to a cheap surrogate oracle) has increased after the NNP optimization step (see Equation 4): $\tilde{E}_{\text{prev}}$ - $\tilde{E}_{\text{cur}} < 0$, where $\tilde{E}_{\text{prev}}$, $\tilde{E}_{\text{cur}}$ are the energies predicted by the surrogate oracle in conformations $s_{\text{cur}}$, $s_{\text{prev}}$ respectively. If this condition is met, the predicted forces in $s_{\text{prev}}$ are considered incorrect and the state $s_{\text{prev}}$ is added to the training dataset. Alternatively, small negative energy changes can be encountered near the local minima due to the nature of gradient-based optimization. We discuss this scenario in Section 4.

## 4 METHOD

Our proposed active learning framework improves on the original GOLF in several aspects. We replace the fixed domain-specific surrogate with a Neural Oracle $g_{\boldsymbol{\phi}}$. The oracle parameters are updated after each training epoch by exponential moving averaging of the online NNP weights $\boldsymbol{\theta}$:

$$\boldsymbol{\phi} = \tau\boldsymbol{\phi} + (1-\tau)\boldsymbol{\theta}. \tag{12}$$

This standard Polyak averaging scheme (Polyak & Juditsky, 1992) is widely used in reinforcement learning (Mnih et al., 2015; Lillicrap et al., 2015; Haarnoja et al., 2018; Guo et al., 2022) and self-supervised learning (Grill et al., 2020) to stabilize the target network. In our work, we utilize this feature to stabilize potential energy estimates and select additional conformations more efficiently.

Moreover, to further optimize active data acquisition, we do not add final conformations of converged (or finished by reaching step limit) trajectories to the training dataset, as the online

---

**Algorithm 1** GOLF with Neural Oracle

---

**Require:** baseline dataset $\mathcal{D}_0$; baseline dataset subsample size $D$; online NNP $f_{\boldsymbol{\theta}}$; neural oracle $g_{\boldsymbol{\phi}}$; genuine oracle $\mathcal{O}_G$; optimizer Opt; optimization step limit $T$; maximum number of negative energy changes $M$; per-cycle additional conformations $k$; total additional conformations $K$; per-cycle epochs $C$; batch size $B$; replay buffer mix ratio $\eta \in [0, 1]$; EMA coefficient $\tau \in [0, 1]$
1: Initialize replay buffer $\mathcal{R} \leftarrow \varnothing$; initialize $\boldsymbol{\phi} \leftarrow \boldsymbol{\theta}$
2: **for** $i \in 1, \ldots, \lceil K/k \rceil$ **do**                      ▷ Runs for $\lceil K/k \rceil$ cycles
3:     $S \leftarrow \varnothing$
4:     **while** $|S| < k$ **do**                      ▷ Additional conformations collection
5:         Sample $s \sim \mathcal{D}$; $t \leftarrow 0$; $m \leftarrow 0$; $\tilde{E}_{\text{prev}} \leftarrow g_{\boldsymbol{\phi}}(s)$; $s_{\text{prev}} \leftarrow s$
6:         **repeat**
7:             $s' \leftarrow s + \text{Opt}\big(\mathbf{F}_{f_{\boldsymbol{\theta}}}(s)\big)$                      ▷ Forces predicted with online NNP
8:             $\tilde{E}_{\text{cur}} \leftarrow g_{\boldsymbol{\phi}}(s')$
9:             **if** $\tilde{E}_{\text{cur}} - \tilde{E}_{\text{prev}} > 0$ **then**                      ▷ Mistake: negative energy change
10:                 $m \leftarrow m + 1$
11:             **end if**
12:             $s_{\text{prev}} \leftarrow s$; $s \leftarrow s'$; $\tilde{E}_{\text{prev}} \leftarrow \tilde{E}_{\text{cur}}$; $t \leftarrow t + 1$
13:         **until** Converged($s$) **or** $t \leq T$ **or** $m < P$   ▷ Optimize until convergence or TL reached
14:         **if** $m = M$ **then**
15:             $S \xleftarrow{\text{add}} s_{\text{prev}}$             ▷ Encountered $M$ mistakes! Add previous conformation to $\mathcal{R}$
16:         **else**
17:             **discard** and resample
18:         **end if**
19:     **end while**
20:     **for all** $s \in S$ **do**                      ▷ Evaluate collected conformations with DFT
21:         $E_{\text{DFT}}(s)$, $\boldsymbol{F}_{\text{DFT}}(s) \leftarrow \mathcal{O}_G(s)$
22:         $\mathcal{R} \xleftarrow{\text{add}} \{s, E_{\text{DFT}}(s), \boldsymbol{F}_{\text{DFT}}(s)\}$
23:     **end for**
24:     Sample shard $\mathcal{D}_i \subset \mathcal{D}_0$ with $|\mathcal{D}_i| = D$                      ▷ Training
25:     **for** $e = 1$ to $C$ **do**
26:         **for** batch $\in \mathcal{D}_i$ **do**
27:             Sample $(1 - \eta)B$ from $\mathcal{D}_j$ and $\eta B$ from $\mathcal{R}$ to form batch $\mathcal{B}$
28:             Update $f_{\boldsymbol{\theta}}$ on $\mathcal{B}$ using Eq. 2,3
29:         **end for**
30:         $\boldsymbol{\phi} \leftarrow \tau \boldsymbol{\phi} + (1 - \tau) \boldsymbol{\theta}$                      ▷ EMA of neural oracle
31:     **end for**
32: **end for**

---

NNP already performs well for such conformations. The GOLF method uses the energy change $\tilde{E}_{\text{prev}} - \tilde{E}_{\text{cur}}$ after the optimization step to select additional conformations. However, due to the nature of the selected optimization algorithm, the energy can sometimes increase near the local minima. This does not necessarily mean that the forces are poorly predicted, but with GOLF's data acquisition scheme, such conformation will still be added to the training dataset, hindering the efficiency of the approach. To minimize the amount of near-optimal non-informative conformations in the training set, we introduce a "mistake budget" $M$. The conformation is only added to the training dataset if there have been $M$ or more negative energy changes in the trajectory.

The training of GOLF Neural Oracle alternates between data collection and supervised updates of the online NNP. In each collection phase, we build a batch $S$ of $k$ conformations. Starting from $s_0 \sim \mathcal{D}$, we iteratively optimize the conformation using prediced forces $\boldsymbol{F}_{\boldsymbol{\theta}}(s)$ and estimate potential energy with a Neural Oracle:

$$s_{\text{cur}} = s_{\text{prev}} + \text{Opt}\big(\boldsymbol{F}_{\boldsymbol{\theta}}(s_{\text{prev}})\big), \qquad \tilde{E}_{\text{cur}} = g_{\boldsymbol{\phi}}(s_{\text{cur}}). \tag{13}$$

We maintain a *mistake counter* $m$ that increments whenever the energy change according to the neural oracle is negative,

$$\tilde{E}_{\text{prev}} - \tilde{E}_{\text{cur}} < 0 \implies m \leftarrow m + 1, \tag{14}$$

Table 1: Evaluation on the test set $\mathcal{D}_{\text{test}}$. Columns report the averaged percentage of minimized energy $\overline{\text{pct}}_{\text{final}}$, the mean residual energy $\overline{E^{\text{res}}}_{\text{final}}$ in kcal/mol, the success rate $\text{pct}_{\text{success}}$ (fraction with $E^{\text{res}} < 1$ kcal/mol), and the diverged conformations percentage $\text{pct}_{\text{div}}$. Arrows indicate preferred direction ($\uparrow$ higher is better; $\downarrow$ lower is better). Best results are in bold.

| Model | $\overline{\text{pct}}_{\text{final}}$ $\uparrow$ (%) | $\overline{E^{\text{res}}}_{\text{final}}$ $\downarrow$ (kcal/mol) | $\text{pct}_{\text{success}}$ $\uparrow$ (%) | $\text{pct}_{\text{div}}$ $\downarrow$ (%) |
|---|---|---|---|---|
| DimeNet++ baseline | 96.21 | 1.67 | 41.45 | 0.85 |
| *Active Learning Models* | | | | |
| Ensemble | 99.99 | 0.001 | 87.6 | 0.25 |
| MVE | 99.86 | 0.096 | 87.4 | 0.85 |
| GOLF–RDKit | 100.13 | -0.04 | 89.05 | 0.55 |
| **GOLF–Neural Oracle (ours)** | **100.24** | **–0.092** | **90.5** | **0.15** |

and we terminate a trajectory on optimizer convergence, a step limit $T$, or when $m = M$. If $m = M$, the previous conformation $s_{\text{prev}}$ is added to $S$. When enough data is collected ($|S| = k$), conformations are evaluated with $\mathcal{O}_G$ and added to the replay buffer $\mathcal{R}$.

After collection, the online NNP $f_{\boldsymbol{\theta}}$ is trained for a small number of epochs using minibatches that mix baseline data from $\mathcal{D}_0$ and additional data $\mathcal{R}$. With mixture coefficient $\eta \in [0, 1]$, each minibatch $\mathcal{B}$ contains a fraction $(1 - \eta)$ drawn from $\mathcal{D}_0$ and a fraction $\eta$ drawn from $\mathcal{R}$. The training objective is defined in Equations 2 and 3. At the end of each epoch, the Neural Oracle is updated via the EMA rule in Equation 12, and the loop returns to the batched collection phase. Over $\lceil K/M \rceil$ cycles, this procedure acquires $K$ DFT-labeled conformations. The full training procedure is detailed in Algorithm 1.

## 5 EXPERIMENTS

### 5.1 $\nabla^2$DFT OPTIMIZATION BENCHMARK

We benchmark the proposed GOLF-Neural Oracle on $\nabla^2$DFT (Khrabrov et al., 2024) benchmark. This benchmark evaluates the optimization performance of NNPs on a subset of 2000 molecules from the $\nabla^2$DFT and utilizes $\omega$B97X-D/def2-SVP level of DFT theory. This benchmark only contains molecules in vacuum, which makes direct comparison with GOLF-RDKit possible.

As baselines, we consider two classical active learning approaches: ensemble and MVE (Tan et al., 2023); the GOLF-RDKit method; and the baseline NNP trained on a fixed dataset without active learning finetuning. The DimeNet++ is used as the NNP for all iterative optimization methods. We use the $\mathcal{D}^{\text{medium}}$ subset of $\nabla^2$DFT to train the baseline NNP and as $\mathcal{D}_0$ for active learning approaches.

For all active learning approaches, we use the same data collection scheme and training hyperparameters except for the conformation selection criterion. The number of additional conformations $K$ equals to 10000. The number of additional conformations per-cycle is $k = 200$, and the number of epochs per cycle is $C = 5$, resulting in a total of 250 training epochs. The initial conformations are sampled from $\mathcal{D}_0$ and then optimized with the L-BFGS (Liu & Nocedal, 1989) algorithm (as implemented in Pytorch Paszke et al. (2019)) using the forces predicted by the NNP (see Equation 13). The training step limit is $T = 100$. We set $\eta = 0.5$, so the batch consists of conformation from $\mathcal{D}_0$ and $\mathcal{R}$ in equal parts.

In the case of the ensemble, the forces are averaged over ensemble members. For the ensemble, we used $\mathcal{M} = 4$ DimeNet++ NNPs. Following (Tan et al., 2023), each NNP was trained on $\mathcal{D}^{\text{medium}}$ using identical hyperparameters, but with different random seeds. The criterion for conformation selection is $\rho_{\text{ensemble}} > t_{\text{ensemble}} = 2.8 \times 10^{-4}$. We found this threshold to strike a good balance between the model's performance and data collection time. For MVE, we modified the final layers of DimeNet++ so that the network outputs the conformational energy along with the per-atom force variance $\sigma^2_{F_i}$. To ensure the positivity of variance values, a soft-plus activation function was applied

Table 2: Ablation on the mistake budget $M$ (maximum negative energy changes per trajectory) and the EMA coefficient $\tau$ used to update the Neural Oracle. Arrows indicate preferred direction.

| Method | $\tau$ | $M$ | $\overline{\text{pct}}_{\text{final}}$ ↑ (%) | $\overline{E^{\text{res}}}_{\text{final}}$ ↓ (kcal/mol) | $\text{pct}_{\text{success}}$ ↑ (%) | $\text{pct}_{\text{div}}$ ↓ (%) |
|---|---|---|---|---|---|---|
| GOLF–RDKit | – | 1 | 100.13 | -0.04 | 89.05 | 0.55 |
| | – | 3 | 100.2 | -0.073 | 89.35 | 0.75 |
| GOLF–Neural Oracle | 0.9 | 1 | 100.11 | -0.035 | 89.0 | 0.3 |
| | 0.9 | 3 | **100.24** | **-0.092** | **90.5** | **0.15** |
| | 0.7 | 3 | 100.18 | -0.061 | 89.5 | 0.4 |
| | 0.5 | 3 | 100.18 | -0.061 | 89.55 | 0.35 |

to the corresponding outputs. The DimeNet++ for MVE was pretrained on $\mathcal{D}_{\text{medium}}$ using loss defined in Equation 10. The same DimeNet++ modification and loss function were used during the active learning phase. The criterion for conformation selection is $\rho_{\text{MVE}} > t_{\text{MVE}} = 1 \times 10^{-5}$.

All iterative optimization baselines featured the same evaluation procedure: the conformations from the $\mathcal{D}^{\text{test}}$ were optimized with L-BFGS until convergence or step limit $T_{\text{eval}} = 200$. The performance on the $\nabla^2$DFT optimization benchmark is in Table 1. The active learning approaches surpass the direct optimization model and the baseline DimeNet++. Moreover, the GOLF approaches outperform the ensemble and MVE. We hypothesize this is because the conformation selection criterion in GOLF merely depends on the model's uncertainty, but instead is specifically tailored for the optimization.

## 5.2 GOLF ABLATIONS

In this section, we study how hyperparameters $\tau$ and $M$ affect the models. The results are provided in Table 2. The "mistake budget" $M > 1$ improves the performance of the GOLF-Neural Oracle, while not significantly affecting the GOLF-RDKit. Despite the fact that energies predicted by MMFF94 (used as a surrogate oracle in GOLF-RDKit) correlate with the DFT energies, the local minima of these functionals differ significantly. Therefore, the MMFF94-predicted energies near the DFT local minima can behave arbitrarily, so the increased $M$ does not help to better select additional training conformations.

For the GOLF-Neural Oracle, $\tau$ regulates the rate of the EMA update (see Equation 12). The lower the update rate, the more Neural Oracle resembles the online NNP, and, conversely, a higher update rate leads to a more stable Neural Oracle. We found that reasonably high values of $\tau$ lead to stable energy estimation and better additional conformation selection.

Table 3: Data-efficient regime. We report the number of additional conformations $K$ used for training and evaluate on $\mathcal{D}_{\text{test}}$ with the same metrics as before. Arrows indicate preferred direction

| Model | $K$ (add. confs) | $\overline{\text{pct}}_{\text{final}}$ ↑ (%) | $\overline{E^{\text{res}}}_{\text{final}}$ ↓ (kcal/mol) | $\text{pct}_{\text{success}}$ ↑ (%) | $\text{pct}_{\text{div}}$ ↓ (%) |
|---|---|---|---|---|---|
| MVE | 1000 | 99.03 | 0.437 | 76.7 | 0.55 |
| **GOLF–Neural Oracle (ours)** | 1000 | **99.19** | **0.376** | **81.45** | **0.45** |

## 5.3 DATA EFFICIENT REGIME

Additionally, we test our proposed method in a "data-efficient" regime, where only 1000 additional conformations are collected during the active learning finetuning. This is especially useful when training with an expensive $\mathcal{O}_G$. For the data efficient regime, we use $K = 1000$, $k = 20$, $C = 1$ and keep other hyperparameters unchanged. The results in Table 3 indicate that our proposed approach remains effective in a data-efficient regime and outperforms other classical active learning approaches.

Table 4: Evaluation on the test set $\mathcal{D}_{\mathrm{SPICE}}^{\mathrm{test}}$. Columns report the averaged percentage of minimized energy $\overline{\mathrm{pct}}_{\mathrm{final}}$, the mean residual energy $\overline{E^{\mathrm{res}}}_{\mathrm{final}}$ in kcal/mol, the success rate $\mathrm{pct}_{\mathrm{success}}$ (fraction with $E^{\mathrm{res}} < 1$ kcal/mol), and the diverged conformations percentage $\mathrm{pct}_{\mathrm{div}}$. Arrows indicate preferred direction ($\uparrow$ higher is better; $\downarrow$ lower is better). Best results are in bold.

| Model | $\overline{\mathrm{pct}}_{\mathrm{final}} \uparrow$ (%) | $\overline{E^{\mathrm{res}}}_{\mathrm{final}} \downarrow$ (kcal/mol) | $\mathrm{pct}_{\mathrm{success}} \uparrow$ (%) | $\mathrm{pct}_{\mathrm{div}} \downarrow$ (%) |
|---|---|---|---|---|
| Nutmeg–Large | 76.60 | 11.17 | 3.67 | 5.45 |
| MACE–OFF23 (Large) | 98.27 | 1.04 | 64.11 | 2.79 |
| **GOLF–Neural Oracle (ours)** | **99.88** | **0.66** | **94.2** | **0.96** |

## 5.4 SPICE

To test both transferability to a more advanced level of DFT theory and to multi-domain chemical data, we benchmark the proposed GOLF-Neural Oracle on the SPICE2.0 dataset. To get the $\mathcal{D}_{\mathrm{SPICE}}^{\mathrm{test}}$, we optimized a subset of test set described in Eastman et al. (2024) (see Appendix A.1 for details). This resulted in $|\mathcal{D}_{\mathrm{SPICE}}^{\mathrm{test}}| = 216$ conformations optimized at $\omega$B97M-D3(BJ)/def2-TZVPPD level of theory. As baselines, we selected the Nutmeg-large model (Eastman et al., 2024) and the MACE-OFF23 (Large) Kovács et al. (2025) as they have already been pretrained on the full SPICE2.0 dataset. We then trained GemNet-OC on the full SPICE dataset and finetuned it with GOLF-Neural Oracle.

As the $\mathcal{D}_{\mathrm{SPICE}}^{\mathrm{test}}$ consists of large ligand molecules, pentapeptides and amino-acid ligand dimers, we filter out molecular systems with less than 40 atoms from the SPICE2.0 dataset and call this filtered dataset $\mathcal{D}_0^{\mathrm{filtered}}$. Hyperparameters specified in Section 5.3 were used to train the GOLF-Neural Oracle on $\mathcal{D}_0^{\mathrm{filtered}}$. The results in Table 4 demonstrate that GOLF-Neural Oracle successfully optimizes almost all molecules from $\mathcal{D}_{\mathrm{SPICE}}^{\mathrm{test}}$, significantly surpassing MACE-OFF23 which was trained without the active learning.

## 6 CONCLUSION

In this work, we have presented a new framework called GOLF with Neural Oracle for molecular conformation optimization learning. We show that a trained neural Oracle can successfully replace a cheap physical simulator, and help the final model achieve a quality comparable with an expensive physical simulator. We thoroughly compare our approach with several baselines, including recent conformation generation models and an adaptation of other active learning schemes. A primary direction for future work is to apply our framework to larger systems and systems dominated by intermolecular interactions, such as optimizing ligand conformations within protein binding pockets or relaxing adsorbates on surfaces. Furthermore, we plan to test the scalability of the Neural Oracle for condensed-phase systems, including liquids and solids under periodic boundary conditions. A particularly promising extension will be the integration of our active learning approach into a hybrid ML/MM (Machine Learning / Molecular Mechanics) framework. This would allow for the data-efficient optimization of a high-accuracy NNP for a reactive site while treating the surrounding environment with a classical force field, enabling the study of chemical processes in large and complex biological systems.

### REPRODUCIBILITY STATEMENT

We will soon release the code used to train all active learning approaches in this study.

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

## A  APPENDIX

### A.1  OPTIMIZATION OF SPICE CONFORMATIONS

To generate the ground truth optimized conformations, we performed full geometry relaxations for each initial conformation from the test set of the SPICE dataset. These calculations were carried out using the Psi4 computational chemistry package (Smith et al., 2020). The optimizations were driven by the optking optimizer, employing the $\omega$B97M-D3BJ (Mardirossian & Head-Gordon, 2016) functional with the def2-TZVPPD (Weigend & Ahlrichs, 2005) basis set to remain consistent with the original level of theory used in the SPICE methodology (Eastman et al., 2024). Each optimization was run until standard convergence criteria were met, defined as the maximum force component on any atom falling below $3 \times 10^{-4}$ Hartree/Bohr, energy change less than $3 \times 10^{-6}$ Hartree, and maximum atom displacement less than $1.2 \times 10^{-3}$ Bohr. Throughout each relaxation, the geometry, potential energy, and interatomic forces were recorded at every step, yielding a complete optimization trajectory for each initial structure.

However, we faced significant challenges during this data generation process, primarily due to the optking optimizer's reliance on an internal coordinate system. For larger and more flexible molecules, the back-transformation from the optimized internal coordinates to Cartesian coordinates frequently failed, leading to a substantial number of unsuccessful optimizations. Consequently, we were only able to generate complete and successful optimization trajectories for approximately one-third of the molecules from the test set. For this successfully optimized subset, a single conformation was relaxed for each molecule to construct our final ground truth dataset.

