# OpenReview forum: "Active Learning for Molecular Conformation Optimization with a Domain-Agnostic Neural Surrogate Oracle"
_ICLR.cc/2026/Conference — ICLR 2026 Conference Withdrawn Submission_

### Official Review · Reviewer_7QE3 · 2025-10-24

**Soundness:** 2
**Presentation:** 2
**Contribution:** 2
**Rating:** 2
**Confidence:** 2

**Summary:**

This paper investigates active learning for molecular conformation optimization, a setting where querying the oracle is time-consuming and the surrogate model trained on collected data may fail to generalize effectively during optimization. The authors propose an active learning framework that maintains two neural network potential (NNP) surrogate models: one learned directly from data, and another updated based on the first model. This approach integrates both conformation optimization and active learning principles. The proposed method is evaluated on publicly available molecular conformation optimization benchmarks.

**Strengths:**

This paper addresses an important problem — active learning for conformation optimization — which is highly relevant to the AI4Science community. The authors present the background and review existing methods clearly, providing readers with a solid foundation to understand the field.

**Weaknesses:**

Although the paper provides a comprehensive introduction, the research question and proposed method are not well-motivated. The rationale for employing two surrogate models is unclear, making it difficult to understand the underlying motivation. Furthermore, the reported improvement appears modest, and the results are presented without error bars. Please refer to my further questions below for more details.

**Questions:**

1. I do not understand the necessity of maintaining two surrogate models — the online NNP and the target NNP — with the target NNP defined as an exponential moving average of the online NNP. This is not clearly explained in Section 4.

2. In Section 4, is the neural oracle equivalent to the target NNP mentioned in the abstract?

3. Could you please include error bars for each experimental result to better assess the statistical significance of the reported improvements?

---

### Official Review · Reviewer_qAh7 · 2025-10-28

**Soundness:** 2
**Presentation:** 3
**Contribution:** 2
**Rating:** 2
**Confidence:** 3

**Summary:**

This work proposes GOLF-Neural Oracle as a follow-up to the GOLF active learning framework, aiming to perform molecular conformational optimization in a more data-efficient way compared with traditional DFT-based approaches. The method achieves state-of-the-art performance in the data-efficient regime.

**Strengths:**

* The motivation to develop the Neural Oracle is clearly presented.
* The method demonstrates superior performance on conformational optimization tasks.

**Weaknesses:**

* In the Introduction, the authors state that methods like MVE, which provide uncertainty estimation, require architectural modifications and retraining, which is considered a burden. However, uncertainty estimation methods such as dropout uncertainty require only minor modifications with no obvious training cost. This raises questions about the motivation for introducing the Neural Oracle.
* The authors claim that the Polyak-averaged Neural Oracle provides more stable potential energy estimates, yet this stability is not well supported by experiments or analytical evidence.
* The paper argues that removing the need to select a surrogate oracle improves applicability to complex domains. However, results are only shown on the SPICE2.0 dataset, without comparison to GOLF on the same data or evaluation on other complex domains.
* The main mechanism of improvement appears to be the introduction of a mistake budget to reduce over-sensitivity to local-minima oscillations. This modification is relatively minor and limits the overall novelty of the method.

**Questions:**

* Please clearly explain why uncertainty-based approaches (e.g., MVE, dropout) are not chosen, and justify the use of exponential moving averaging with either analytical reasoning or experimental evidence.
* Please justify why the Neural Oracle is expected to generalize well to broader chemical domains beyond those tested.
* Kindly add an LLM usage disclosure section in the appendix, as required by ICLR policy.

---

### Official Review · Reviewer_mdhw · 2025-10-31

**Soundness:** 2
**Presentation:** 1
**Contribution:** 1
**Rating:** 2
**Confidence:** 3

**Summary:**

The paper proposes a new active learning method for molecular conformation optimization. Instead of using a fixed empirical surrogate oracle like in prior work (e.g., GOLF with MMFF94), it uses a learnable neural oracle updated via exponential moving average (EMA). The method is model-agnostic, simple to implement, and avoids the cost of training ensembles or changing model architecture.

**Strengths:**

1. The main idea of replacing the empirical surrogate with an EMA-based neural oracle is simple and general. It avoids the need for domain-specific tuning.

2. The method is compatible with existing NNP models and does not require any changes to their architecture.

3. The empirical results are strong, and the method performs well even when only 1000 additional samples are used.

**Weaknesses:**

1. The paper does not provide a detailed explanation of why EMA leads to stable uncertainty estimation. The justification appears to be mainly empirical.

2. The criterion used for selecting conformations, which depends on counting negative energy changes, feels somewhat heuristic.

3. In the SPICE experiments, the proposed method benefits from finetuning, while the baseline models are used as-is. This may affect the fairness of the comparison.

**Questions:**

1. What motivated the choice of EMA over other methods for estimating uncertainty, such as dropout-based approaches or evidential models?

2. Can this approach be extended to highly flexible systems, such as large biomolecules, where energy estimates may become less reliable due to structural noise?

---

### Official Review · Reviewer_NCBe · 2025-10-31

**Soundness:** 3
**Presentation:** 3
**Contribution:** 2
**Rating:** 4
**Confidence:** 3

**Summary:**

This paper introduces GOLF-Neural Oracle, an active learning approach for molecular conformation optimization. Building off the GOLF framework, the authors train a NNP to replace the traditional force field in GOLF. During training, queries are sent to the DFT oracle to get high-quality datapoints to train the model. Results show that the proposed approach outperforms GOLF with RDKit force fields, as well as a set of other baselines both with and without active learning.

**Strengths:**

1. The motivation for introducing a NNP to replace an empirical force field is quite reasonable, since as the authors say there are many interesting systems for which traditional force fields are not accurate enough (especially in materials science)
2. The proposed framework is quite well thought-out, and addresses the problem. The target network weighted average technique is interesting and seems to perform well
3. The results comparing GOLF-Neural Oracle with baselines, especially in Tables 3 and 4, are convincing and quite strong

**Weaknesses:**

The main weakness in my opinion is the novelty compared to the GOLF framework. The main difference is the Neural Oracle, which while interesting, is not necessarily a major methodological novelty. The other modification, the mistake budget M, is well-motivated but I'm not sure how much its use makes a difference. Looking at the tables that compare GOLF-RDKit with GOLF-Neural Oracle (Tables 1 and 2), I don't really see a major improvement in the introduced method vs GOLF-RDKit. While there does seem to be a small benefit, it's usually only by 1 or a few percentage points. Given the simplicity of the modifications, I would only find this paper very interesting if the modifications led to very significant improvements over GOLF, which I'm not sure I see.

While the comparisons to other methods in Tables 3 and 4 are strong, I think the major baseline is GOLF-RDKit, and all the comparisons with that method yield very limited benefit for GOLF-Neural Oracle.

**Questions:**

N/A

---

### Official Review · Reviewer_Yp4T · 2025-11-02

**Soundness:** 2
**Presentation:** 3
**Contribution:** 2
**Rating:** 4
**Confidence:** 3

**Summary:**

This paper proposes GOLF–Neural Oracle, a new active learning framework for training neural network potentials (NNPs) in molecular conformation optimization.
The method removes the dependency on empirical surrogate force fields (such as MMFF94 used in GOLF) by introducing a trainable surrogate oracle — a Neural Oracle updated as an exponential moving average (EMA) of the online NNP.
The authors benchmark their approach on the $\nabla^2$DFT and SPICE2.0 datasets, showing consistent improvements over baselines (Ensemble, MVE, GOLF-RDKit) in both standard and data-efficient regimes.

**Strengths:**

* Clear motivation and problem formulation.

* Using an EMA-updated Neural Oracle is an appealing and lightweight idea inspired by BYOL. It avoids ensemble training, uncertainty prediction heads, or architectural modifications, making it broadly applicable to existing NNP pipelines.

* The results (Tables 1–4) consistently show that GOLF-Neural Oracle achieves the best or comparable performance with fewer additional conformations.

* The method can be plugged into any molecular NNP training pipeline to improve data efficiency, which may be valuable for high-cost DFT or ab initio workflows.

**Weaknesses:**

* The analogy to BYOL is intuitive, but the paper never explains why EMA-averaged weights provide a meaningful uncertainty signal or how this leads to improved sampling.
A theoretical or empirical calibration study (e.g., correlation between oracle energy error and true uncertainty) is missing.

* The experiments focus solely on end-metrics but do not analyze uncertainty quality or convergence behavior during active learning cycles. It is difficult to verify whether improvements stem from better uncertainty estimation or simply additional training dynamics.

* The “domain-agnostic” claim is overstated to some extent. All benchmarks involve isolated molecules in vacuum or small solvated systems.
No results are shown for periodic, condensed-phase, or large biomolecular systems — the domains where empirical force fields indeed fail.

* No runtime or computational-cost evaluation.

**Questions:**

* How does the method scale to larger molecules systems?

* How sensitive is performance to $\tau$ and M across different datasets — are these hyperparameters transferable?

---

### Note · Authors · 2025-11-28

I have read and agree with the venue's withdrawal policy on behalf of myself and my co-authors.